# Using Hierarchies to Efficiently Combine Evidence with Dempster's Rule of Combination

**Daira Pinto Prieto**[1]                    **Ronald de Haan**[1]

[1]Institute for Logic, Language and Computation, University of Amsterdam, The Netherlands

## Abstract

Dempster's rule of combination allows us to combine various independent pieces of evidence that each have a certain degree of uncertainty. This provides a useful way for dealing with uncertain evidence, but the rule is computationally intractable. In this paper, we analyze the complexity of this rule for differently structured bodies of evidence and we consider a known algorithm by Shafer and Logan to compute this rule efficiently over a hierarchical set of evidence. We show that one can check in polynomial time whether an arbitrary set of evidence has a hierarchical shape, enabling the use of Shafer and Logan's algorithm. Moreover, we consider two different approaches to deal with non-hierarchical sets of evidence: (i) considering hierarchical subsets and (ii) taking advantage of internal hierarchical structures in the overall set. For the former case, we conclude that getting different hierarchies from an arbitrary set of pieces of evidence corresponds to the VERTEX-COVER problem and we present algorithms for obtaining these hierarchies based on this correspondence. For the latter case, we present a fixed-parameter tractable algorithm which computes the belief function of any piece of evidence included in the set.

## 1 INTRODUCTION

Dempster-Shafer Theory (DST) [Shafer, 1976, Yager and Liu, 2008, Liu and Yager, 2008] provides a toolbox for modelling evidence with different forms of uncertainty and for merging such evidence. The theory originated in the context of inferential statistics, where there are other widely used methods for modelling evidence, such as Bayesian statistics. An important technical property of DST is that it rejects the additivity principle, which lies at the basis of

probability theory: given two disjoints events $A$ and $B$, the likelihood assigned to $A \cup B$ is not required to be equal to that of $A$ plus that of $B$—but greater or equal. This aspect is closely tied to the following two facts: (i) DST is a direct generalization of Bayesian probability theory, and (ii) DST provides an expressive and powerful way of representing ignorance.

In particular, one can directly express for which propositions any piece of evidence provides support, and for which propositions this piece of evidence *does not* provide support. This explicit representation of ignorance is in strong contrast to the case of Bayesian probability, where evidence *for* some proposition $p$ is, by design, evidence *against* its complement $\neg p$. DST has been argued to be particularly useful in cases where incomplete evidence plays an important role—see, e.g., [Barnett, 1981].

We can illustrate these properties with a differential diagnosis of having a red eye (Table 1). According to this symptom, a physician can establish a universe of five possible causes: *hyposphagmus* ($H$), *acute conjunctivitis* ($C$), *acute glaucoma* ($G$), *keratitis* ($K$) and *anterior uveitis* ($U$). [1] Then, if the patient reports to have pain in the eyes, in DST, some value of likelihood can be associated to $\{G, K, U\}$. However, this evidence does not point towards any of the three individual possibilities $G$, $K$ and $U$. In addition, if the patient tells that she does not know whether she has lower visual acuity, in DST, we can express that this is very weak

|  | $H$ | $C$ | $G$ | $K$ | $U$ |
|---|---|---|---|---|---|
| Pain | no | no | yes | yes | yes |
| Foreign body sens. | no | yes | no | yes | no |
| Pupil size | norm. | norm. | mydr. | mios. | mios. |
| Lower visual acuity | no | no | yes | no | yes |

Table 1: Background knowledge about eye diseases

[1]Here, we make the assumption that *exactly* one of these causes the symptom.

*Accepted for the 38th Conference on Uncertainty in Artificial Intelligence* (UAI 2022).

evidence both against $H$, $C$ and $K$, and against $G$ and $U$ without considering it as (strong) evidence *for* any proposition.

Given these theoretical advantages for representing uncertainty, DST has been used to combine human opinions [Altieri et al., 2017, Agreh and Ghaffari-Hadigheh, 2019] or to merge data [Wu et al., 2002, 2003], for example. However, the disadvantage is that combining evidence in DST—using Dempster's rule of combination—is #P-hard [Orponen, 1990], presenting an obstacle for wider use of DST. Some results have been established on restricted cases where evidence can be combined in a computationally efficient way [Barnett, 1981, Shafer and Logan, 2008, Shafer et al., 1987, Bergsten and Schubert, 1993], but a structured computational complexity analysis has been missing from the literature.

**Contributions** In this paper, we address this gap by initiating a structured and detailed complexity analysis of using Dempster's rule of combination. In particular, we show that the problem remains #P-hard when restricted to simple support functions, and when restricted to evidence for propositions containing limited information (i.e., the set representing the proposition is nearly the entire universe). We also point out some cases that admit polynomial-time algorithms—most prominently the case where the propositions for (or against) which we have evidence form a hierarchical structure (in this case, a polynomial-time algorithm by Shafer and Logan [2008] can be used). We give a polynomial-time algorithm for deciding when such hierarchical structure is present. Moreover, we analyze the complexity of several problems related to removing some pieces of evidence in order to obtain a hierarchy structure. We show that these problems are NP-complete, but can be solved in fixed-parameter tractable time in cases where only a few pieces of evidence need to be removed. Finally, we generalize the algorithm of Shafer and Logan to arbitrary sets, and we show that this generalized algorithm runs in fixed-parameter tractable time when parameterized by a certain notion of how much hierarchical structure is present in the evidence.

**Outline** First, we will recall some definitions of Dempster-Shafer Theory (in Section 2). We will provide a complexity analysis of the problem of using Dempster's rule in Section 3. Then, in Section 4, we will study the problem of deciding if a given body of evidence forms a hierarchy. Sections 5 and 6 are reserved to explore different possibilities for combining evidence that does not form a hierarchy.

Due to space reasons, for some statements in the paper, we defer the proof to the appendix—we use the symbol $\star$ to indicate this.

## 2 PRELIMINARIES

In this section, we will give a brief overview of the main technical elements appearing in Dempster-Shafer Theory (DST). The theory revolves around *belief functions*, which assigns to each proposition—i.e., each subset of possible states—a degree of belief. Moreover, the main focus is on belief functions that are supported by objective pieces of evidence—these are called *support functions*. The atomic building blocks to construct support functions are called *simple support functions*, which express evidence for a single proposition, and which can be combined using Dempster's rule of combination into more complex support functions. And the other way around, any support function can be decomposed into simple support functions [Smets, 1995].

**Definition 2.1.** *The* frame $\Theta$ of discernment *is the set of all different possible states in a given context. A* basic probability assignment (b.p.a.) *over the frame $\Theta$ is a function $m : 2^\Theta \to [0, 1]$ such that $m(\emptyset) = 0$ and $\sum_{A \subseteq \Theta} m(A) = 1$.*

**Definition 2.2** (Focal element)**.** *Given a frame of discernment $\Theta$ and a b.p.a. $m$, a subset $A \subseteq \Theta$ is a* focal element *of $m$ if $m(A) > 0$. Any focal element $A \neq \Theta$ we will call a* proper focal element*. In addition, we will say that two focal elements $A, B \subseteq \Theta$ are* complementary *if $A = \overline{B} = \Theta \setminus B$.*

**Definition 2.3** (Simple b.p.a.)**.** *A b.p.a. is called* simple *if it has exactly one proper focal element $A$. That is, if $m(A) > 0$, and for all $B \in 2^\Theta \setminus \{A, \Theta\}$ it holds that $m(B) = 0$.*

**Definition 2.4** (Dempster's rule of combination; DRC)**.** *Let $m_1$ and $m_2$ be b.p.a.'s over the same frame $\Theta$ of discernment with focal elements $A_1, \ldots, A_k$ and $B_1, \ldots, B_\ell$, respectively. Moreover, suppose that $\sum_{A_i \cap B_j = \emptyset} m_1(A_i) m_2(B_j) < 1$. Then the following b.p.a. $m$, also denoted by $m_1 \oplus m_2$, is the result of applying Dempster's rule of combination to $m_1$ and $m_2$: $m(\emptyset) = 0$ and $m(C) = \sum_{A_i \cap B_j = C} m_1(A_i) m_2(B_j)/K$, where $K$ is the normalization factor $1 - \sum_{A_i \cap B_j = \emptyset} m_1(A_i) m_2(B_j)$, for all nonempty sets $C \subseteq \Theta$.*

Basic probability assignments provide a useful way of representing all the evidence that is available in a given situation, but one is typically interested in summary measures such as belief and plausibility functions.

**Definition 2.5** (Belief and plausibility)**.** *Let $\Theta$ be a frame of discernment, and let $m : 2^\Theta \to [0, 1]$ be a basic probability assignment. Then $\mathrm{Bel}_m : 2^\Theta \to [0, 1]$ is called the* belief *based on $m$, and is defined by letting $\mathrm{Bel}_m(A) = \sum_{B \subseteq A} m(B)$ for each $A \subseteq \Theta$. Moreover, $\mathrm{Plau}_m : 2^\Theta \to [0, 1]$ is called the* plausibility *based on $m$, and is defined by letting $\mathrm{Plau}_m(A) = \sum_{B \subseteq \Theta, B \cap A \neq \emptyset} m(B)$. As a result, for each $A \subseteq \Theta$ it holds that $\mathrm{Plau}_m(A) = 1 - \mathrm{Bel}_m(\overline{A})$ and that $\mathrm{Bel}_m(A) \leq \mathrm{Plau}_m(A)$. When $m$ is clear from the context, we will write $\mathrm{Bel}$ to denote $\mathrm{Bel}_m$ and $\mathrm{Plau}$ to denote $\mathrm{Plau}_m$.*

In the remainder of the paper, we will focus on applying DRC to simple support functions and *dichotomous support functions*—which have two proper focal elements that are complementary—and we will assume that separate pieces of evidence are given in the form of (simple) b.p.a.'s $m_1, \ldots, m_n$ over some finite frame $\Theta$ of discernment, which are combined by using DRC into $m = \bigoplus_{i=1}^n m_i$.

# 3 THE COMPLEXITY OF USING DEMPSTER'S RULE OF COMBINATION

One of the main obstacles for using Dempster's rule of combination (DRC) is its high computational complexity. Orponen proved that computing beliefs based on the application of DRC to arbitrary support functions is #P-hard by giving a reduction from the well-known #SAT problem for CNF formulas [Orponen, 1990]. In this section, we will provide a complexity analysis that builds forth on Orponen's hardness result. In particular, we will identify various restrictions under which the problem remains #P-hard, and we will identify some restrictions that allow polynomial-time algorithms.

## 3.1 HARDNESS FOR SIMPLE SUPPORT FUNCTIONS

Throughout the paper, we make the assumption that the individual pieces of evidence are all given as simple support functions—that is, support functions with a single proper focal element. In principle, it could be the case that the hardness result of Orponen does not apply to the case where DRC is only applied to simple support functions—as the reduction used to show hardness involves belief functions with multiple proper focal elements. We begin with showing that, in fact, computing beliefs based on the application of DRC to simple support functions is #P-hard.

The reduction that we give is very much similar to the reduction given by Orponen [1990, Theorem 3.1]—the main difference being that we take a restricted variant of #SAT to reduce from. Nevertheless, we give a description of this reduction—to make this paper self-contained, allowing the reader to understand and appreciate the various complexity results in this section that are based on (variations of) this proof. Moreover, our presentation of (the correctness argument in) the proof differs from that of Orponen, providing the reader with another entry into understanding the argument.

We begin with laying out the precise statements of two computational problems related to the application of DRC. The difference between these to problems is whether the required output is (1) the mass value $m(A)$ or (2) the belief $\mathrm{Bel}_m(A)$—both based on the combined mass function $m = \bigoplus_{i=1}^n m_i$.

DRC-COMPUTE-MASS
*Input:* A frame of discernment $\Theta$, b.p.a.'s $m_1, \ldots, m_n$ over $\Theta$, and a set $A \subseteq \Theta$.
*Output:* $m(A) = (\bigoplus_{i=1}^n m_i)(A)$.

DRC-COMPUTE-BELIEF
*Input:* A frame of discernment $\Theta$, b.p.a.'s $m_1, \ldots, m_n$ over $\Theta$, and a set $A \subseteq \Theta$.
*Output:* $\mathrm{Bel}_m(A)$ for $m = \bigoplus_{i=1}^n m_i$.

Both variants of the problem are #P-hard, even when restricted to simple support functions.

**Theorem 3.1.** DRC-COMPUTE-MASS *is #P-complete. Moreover, #P-hardness holds even when restricted to the case where $m_1, \ldots, m_n$ are simple b.p.a.'s and $|A| = 1$.*

*Proof (sketch).* We show #P-hardness by providing a reduction from the #P-complete problem #MON-SAT, which concerns counting the number of satisfying truth assignments of a propositional CNF formula that is monotone (variables occur only positively) [Valiant, 1979]. We reiterate that this reduction is entirely similar to the reduction used to show #P-hardness of DRC-COMPUTE-MASS in general [Orponen, 1990]—i.e., without the restriction to *simple* support functions.

Let $\varphi = c_1 \wedge \cdots \wedge c_k$ be a monotone propositional CNF formula over the variables $x_1, \ldots, x_n$. We define $\Theta = \{1, \ldots, k, *\}$ and $A = \{*\}$, and we construct $n$ simple basic probability assignments $m_1, \ldots, m_n$. Each b.p.a. $m_i$ has as single focal element $T_i = \{*\} \cup \{ j \mid \text{clause } c_j \text{ does not contain literal } x_i \}$ where:

$$m_i(T_i) = m_i(\Theta) = 1/2, \text{ and}$$
$$m_i(B) = 0 \text{ for each } B \in 2^\Theta \setminus \{T_i, \Theta\}.$$

Now, let $m = \bigoplus_{i=1}^n m_i$. We will show that $m(A) = m(\{*\}) = q 2^{-n}$, where $q$ is the number of satisfying truth assignments of $\varphi$. Firstly, observe that each $m_i$ assigns non-zero mass only to sets that include $*$, and therefore there is no sequence of sets in the Cartesian product $\boldsymbol{A} = \times_{i=1}^n \{T_i, \Theta\}$ that has an empty intersection. As a result, we get that $m(A)$ equals the sum of $\prod_{i=1}^n m_i(A_i)$ for all sequences $(A_1, \ldots, A_n) \in \boldsymbol{A}$ such that $\bigcap_{i=1}^n A_i = \{*\}$. Moreover, for each sequence $(A_1, \ldots, A_n) \in \boldsymbol{A}$, it holds that $\prod_{i=1}^n m_i(A_i) = 2^{-n}$.

Consider the following bijection $\sigma$ between truth assignments $\alpha : \{x_1, \ldots, x_n\} \rightarrow \{0, 1\}$ and sequences $(A_1, \ldots, A_n) \in \boldsymbol{A}$, where for each $\alpha$, we let $\sigma(\alpha) = (A_1, \ldots, A_n)$ such that $A_i = T_i$ if $\alpha(x_i) = 1$, and $A_i = \Theta$ if $\alpha(x_i) = 0$.

We argue that for each $\alpha : \{x_1, \ldots, x_n\} \to \{0, 1\}$ it holds that $\alpha$ satisfies $\varphi$ if and only if, for $\sigma(\alpha) = (A_1, \ldots, A_n)$ it holds that $\bigcap_{i=1}^{n} A_i = \{*\}$. This suffices to show that $m(A) = q2^{-n}$. $\qquad\square$

Note that in the proof of Theorem 3.1 we can define the b.p.a.'s $m_i$ with only a single proper focal element for the exact reason that $\varphi$ is monotone. If $\varphi$ were not monotone, we would have to assign a non-zero mass to the sets $F_i = \{*\} \cup \{ j \mid \text{clause } c_j \text{ does not contain literal } \neg x_i \}$ to make the reduction work, as in the original #P-hardness proof for DRC-COMPUTE-MASS [Orponen, 1990]. Put differently, due to the fact that $\varphi$ is monotone, we get that $F_i = \Theta$.

Now, because for any singleton set $A$ it holds that $m(A) = \text{Bel}_m(A)$, we can conclude #P-hardness also for the problem DRC-COMPUTE-BELIEF under the same restrictions.

## 3.2 SIZE BOUNDS ON FOCAL ELEMENTS

In the previous subsection, we saw that restricting our attention to simple support functions is not enough to guarantee that we can use DRC in polynomial time. In this subsection, we will consider two additional restrictions—both based on additional constraints on the size of (proper) focal elements.

The first additional restriction that we consider is that the size of the proper focal elements of the belief functions are bounded by some fixed constant. This restriction allows us to use DRC in polynomial time.

**Proposition$^\star$ 3.2.** *Let $c \in \mathbb{N}$ be a fixed constant. DRC-COMPUTE-MASS and DRC-COMPUTE-BELIEF can be computed in polynomial time if the b.p.a.'s $m_1, \ldots, m_n$ only have proper focal elements of size $\leq c$.*

Restricting proper focal elements to be of bounded size corresponds to the requirement that all pieces of evidence are highly informative—or in other words, that each piece of evidence assigns zero mass to all but a few possibilities. Arguably, this occurs only in a very limited set of circumstances, limiting the practical value of the tractability result of Proposition 3.2. Moreover, it has been argued that in various circumstances—e.g., in cases where the different pieces of evidence are to a large extent contradictory—DRC might not be the most appropriate way to combine evidence (see, e.g., Yager, 1987, Pearl, 1990, Jøsang and Pope, 2012).

Another restriction, that is perhaps more promising for practical applications, consists of restricting pieces of evidence to be of limited informativeness: allowing only simple support functions whose single proper focal element consists of $\Theta$ with only a constant number of possibilities removed. Unfortunately, this restriction does not alleviate the computational intractability of using DRC.

**Proposition 3.3.** *Let $c \geq 3$ be a fixed constant. DRC-COMPUTE-MASS and DRC-COMPUTE-BELIEF are #P-*

*hard even when restricted to the case where the b.p.a.'s $m_1, \ldots, m_n$ all have a single proper focal element that is of size $\geq |\Theta| - c$, and where $|A| = 1$.*

*Proof (sketch).* Similarly to the proof of Theorem 3.1, we adapt the reduction by Orponen [1990]. This time, we take as starting point for the reduction the restriction of #MON-SAT where each variable appears in at most 3 clauses. The problem remains #P-complete under this restriction [Greenhill, 2000]. The resulting instance then has the property that each of $m_1, \ldots, m_n$ has a single proper focal element that is of size $\geq |\Theta| - c$, and that $|A| = 1$. $\quad\square$

## 3.3 HIERARCHICALLY STRUCTURED FOCAL ELEMENTS

Nevertheless, there are some cases where it is possible to avoid the worst-case computational complexity of using DRC. Concretely, Shafer and Logan [2008] proved that given a hierarchical structure of focal elements, we can compute the total belief, commonality and plausibility of certain sets in polynomial time.

In this subsection, we will introduce this algorithm, and in Sections 4 and 5, we will investigate how to decide if this algorithm can be used to efficiently aggregate (a subset of) the evidence in a given situation. Moreover, in Section 6, we will study a way to extend this algorithm to arbitrary sets of evidence.

**Definition 3.4** (Hierarchy). *Let $\Theta$ be a frame of discernment. A set $\mathcal{H} = \{A_1, \ldots, A_m\}$ of focal elements $A_i \subseteq \Theta$ is a* hierarchy *over $\Theta$ if there exists a tree where the root node is labelled with $\Theta$, and all other nodes are labelled with an element $A_i \in \mathcal{H}$ such that: (1) if a node labelled with $A_j$ is the child of a node labelled with $A_i$, then $A_j \subseteq A_i$, and (2) if two nodes labelled with $A_i$ and $A_j$ are siblings, then $A_i \cap A_J = \emptyset$. In other words, a set $\mathcal{H} = \{A_1, \ldots, A_m\} \subseteq 2^\Theta$ is a hierarchy if and only if for all $A_i$ and $A_j \in \mathcal{H}$ it holds that: if $A_i \cap A_j \neq \emptyset$, then $A_j \subseteq A_i$ or $A_i \subseteq A_j$.*

**Example 3.5.** *Consider $\Theta = \{a, b, c, d, e\}$. Then $\mathcal{H} = \{A_1, \ldots, A_6\}$, for $A_1 = \{a, b, c\}$, $A_2 = \{d, e\}$, $A_3 = \{a, b\}$, $A_4 = \{a\}$, $A_5 = \{e\}$ and $A_6 = \{d\}$, is a hierarchy. A corresponding tree is shown on the right.*

**Theorem 3.6** (Shafer and Logan, 2008). *Given a hierarchy $\mathcal{H}$ and given simple b.p.a.'s $m_1, \ldots, m_n$ each of whose single proper focal element is either an element of $\mathcal{H}$ or the complement of one, then $\text{Bel}_m(A)$, $\text{Bel}_m(\overline{A})$, $\text{Plau}_m(A)$ and $\text{Plau}_m(\overline{A})$, for $m = \bigoplus_{i=1}^{n} m_i$, can be computed in polynomial time for all $A \in \mathcal{H}$.*

The algorithm provided by Theorem 3.6 does not allow us to efficiently compute the belief (or plausibility) of arbitrary sets $A \subseteq \Theta$, but only for sets $A \in \mathcal{H}$ (or their complements). Nevertheless, it is plausible to assume that if the available evidence in a given application domain forms a hierarchy (as specified in Definition 3.4), then sets $A \subseteq \Theta$ of interest also match this hierarchy. This is the case, for example, in the setting of diagnostic reasoning in medicine, where hierarchical evidence naturally appears [Gordon and Shortliffe, 2008].

# 4 PLACING ALL EVIDENCE IN A HIERARCHY

Suppose that a researcher has performed different experiments that provide evidence for various subsets of a frame of discernment $\Theta$ and is interested in knowing which elements of $\Theta$ have the most support according to it.

It would be very useful to know whether these focal elements form a hierarchy, in which case they could use the Shafer-Logan algorithm (Theorem 3.6) to efficiently compute beliefs based on the combination of all the evidence.

In this section, we will study the problem of deciding if a given set of evidence is in line with a single hierarchy. The Shafer-Logan algorithm works for the case where for each set $A$ in the hierarchy a dichotomous b.p.a. is given—which assigns weight to $A$ and to its complement $\Theta \setminus A$. Therefore, we take as starting point for this problem a set $\mathcal{A}$ with $m$ pairs of sets, each consisting of a focal element and its complement—one per each dichotomous b.p.a.

In particular, we will give a polynomial-time algorithm that decides, given a set $\mathcal{A}$ of such pairs of sets, whether we can form a hierarchy $\mathcal{H}$ by taking exactly one set from each pair—in which case we say that $\mathcal{A}$ *admits a hierarchy*—and that computes what this hierarchy would look like.

**Definition 4.1** (Conflict). *We will say that there exists a* conflict *between two focal elements $A_i$ and $A_j$ if $A_i \not\subseteq A_j$, $A_j \not\subseteq A_i$, and $A_i \cap A_j \neq \emptyset$.*

*We will denote such a conflict with $A_i \rightleftharpoons A_j$.*

**Theorem 4.2.** *Let $\Theta$ be a frame of discernment. Given a set $\mathcal{A} = \{(B_i, \overline{B_i})\}_{i=1}^{m}$ of pairs, where for each $1 \leq i \leq m$, $B_i$ and $\overline{B_i} = \Theta \setminus B_i$ are complementary sets over $\Theta$, the following are equivalent:*

(a) *There exists a hierarchy $\mathcal{H}$ consisting of exactly one set from each pair in $\mathcal{A}$.*

(b) *There exists a set $\{A_1, \ldots, A_m\}$ of focal elements formed by exactly one element of each pair in $\mathcal{A}$ such that for each two distinct $A_i, A_j \in \{A_1, \ldots, A_m\}$ it holds that $A_i \not\rightleftharpoons A_j$.*

(c) *The following propositional 2CNF formula $\varphi = \varphi_1 \wedge \varphi_2$ over the variables $x_1, \ldots, x_m, y_1, \ldots, y_m$*

*is satisfiable. For each $1 \leq i \leq m$, let $\nu(B_i) = x_i$ and $\nu(\overline{B_i}) = y_i$. Then $\varphi_1$ consists of the clauses $(\nu(B_i) \vee \nu(\overline{B_i}))$ and $(\neg \nu(B_i) \vee \neg \nu(\overline{B_i}))$ for each $1 \leq i \leq m$, and $\varphi_2$ consists of the clauses $(\neg \nu(A_i) \vee \neg \nu(A_j))$ for each $A_i \in \{B_i, \overline{B_i}\}$ and $A_j \in \{B_j, \overline{B_j}\}$ such that $A_i \rightleftharpoons A_j$.*

*Proof.* One can straightforwardly show that (a) and (b) are equivalent by using Definitions 3.4 and 4.1. For space reasons, we omit further details on this.

We then show that (b) implies (c). Suppose that there is a set $\{A_1, \ldots, A_m\}$ of focal elements formed by exactly one element of each pair in $\mathcal{A}$ such that for each two distinct $A_i, A_j \in \{A_1, \ldots, A_m\}$ it holds that $A_i \not\rightleftharpoons A_j$. We then define a truth assignment $\alpha$ that satisfies $\varphi$. For each $1 \leq i \leq m$, let $\alpha(x_i) = 1$ and $\alpha(y_i) = 0$ if $B_i \in \{A_1, \ldots, A_m\}$ and let $\alpha(x_i) = 0$ and $\alpha(y_i) = 1$ if $B_i \notin \{A_1, \ldots, A_m\}$. This assignment satisfies $\varphi_1$ because for each $1 \leq i \leq m$ there is exactly one of $B_i, \overline{B_i}$ in the set $\{A_1, \ldots, A_m\}$. The clauses in $\varphi_2$ are also satisfied by $\alpha$ because there are no two distinct $A_i, A_j \in \{A_1, \ldots, A_m\}$ such that $A_i \rightleftharpoons A_j$.

Finally, we show that (c) implies (b). Take a truth assignment $\alpha$ that satisfies $\varphi$. For each $1 \leq i \leq m$, we let $A_i = B_i$ if $\alpha(x_i) = 1$ and $A_i = \overline{B_i}$ if $\alpha(x_i) = 0$. Then $\{A_1, \ldots, A_m\}$ contains exactly one element of each pair in $\mathcal{A}$. Now, to derive a contradiction, suppose that there were two distinct $A_i, A_j \in \{A_1, \ldots, A_m\}$ with $A_i \rightleftharpoons A_j$. Then $\varphi$ would contain the clause $(\neg \nu(A_i) \vee \neg \nu(A_j))$, and $\alpha(\nu(A_i)) = \alpha(\nu(A_j)) = 1$, and so $\alpha$ would not satisfy $\varphi$, which contradicts our assumption. Therefore, we can conclude that there are no two distinct $A_i, A_j \in \{A_1, \ldots, A_m\}$ such that $A_i \rightleftharpoons A_j$. $\square$

In fact, the proof of Theorem 4.2 also shows that we can efficiently construct a hierarchy $\mathcal{H}$ containing exactly one element from each pair in $\mathcal{A}$ from a truth assignment for the 2CNF formula.

**Corollary 4.3.** *Let $\Theta$ be a frame of discernment and $\mathcal{A}$ be a set of complementary focal elements. Moreover, let $\varphi$ be the 2CNF formula described in Theorem 4.2. Then from any satisfying assignment $\alpha$ for $\varphi$, we can in polynomial time construct a hierarchy $\mathcal{H}$ containing exactly one element from each pair in $\mathcal{A}$.*

Since one can in linear time decide whether a given 2CNF formula is satisfiable (and if so, find a satisfying truth assignment) [Aspvall et al., 1979], we can in polynomial time decide whether the pairs in $\mathcal{A}$ admit a hierarchy, and compute such a hierarchy if this is the case.

# 5 PLACING AS MUCH EVIDENCE AS POSSIBLE IN A HIERARCHY

In the previous section we studied how to efficiently decide if a given set of evidence can all be placed in a single hierarchy. Of course, this is not always possible. In this section, we will study algorithms to form hierarchies that accommodate much (but not all) of a given set $\mathcal{A}$ of evidence.

Again, as in the previous section, we suppose that we are given a set $\mathcal{A}$ with $m$ pairs $(B_i, \overline{B}_i)$ of sets, each consisting of a focal element and its complement—with the underlying idea that we have evidence in the form of simple b.p.a.'s whose proper focal elements are $A_i$ or $\overline{A}_i$. Then, if there is no single hierarchy in line with all of this evidence, we can only obtain a hierarchy by selecting one set from some (but not all) pairs $(B_i, \overline{B}_i)$.

One way to make such a selection is the following. We take the 2CNF formula $\varphi$ from Theorem 4.2, and adapt it into $\varphi' = \varphi'_1 \wedge \varphi_2$, where $\varphi'_1$ consists only of the clauses $(\neg\nu(B_i) \vee \neg\nu(\overline{B}_i))$. Then, any satisfying truth assignment for $\varphi'$ corresponds to a hierarchy that fits a subset of the evidence. However, by taking this crude approach, we have no influence on how much of the evidence is accommodated by the resulting hierarchy—for example, one can satisfy $\varphi'$ by setting all variables to false, which corresponds to the trivial, empty hierarchy.

## 5.1 STRUCTURE IN THE SET OF CONFLICTS

We will start with distinguishing some structure in the set of conflicts between focal elements, that will turn out to be useful to develop algorithms for finding (large) partial hierarchies. In the remainder, we will assume that a set $\mathcal{A}$ with $m$ pairs $P_i = (B_i, \overline{B}_i)$ of complementary sets is given.

**Definition 5.1** (Conflict between pairs of focal elements). *Let $P_1 = (A_1, \overline{A}_1)$ and $P_2 = (A_2, \overline{A}_2)$ be two pairs of complementary sets. Moreover, let $\ell$ be the number of conflicts between the sets appearing in $P_1$ and $P_2$—that is, $\ell = |\{(B_1, B_2) \mid B_1 \in \{A_1, \overline{A}_1\}, B_2 \in \{A_2, \overline{A}_2\}, B_1 \rightleftharpoons B_2\}|$. We then say that there are $\ell$ conflicts between $P_1$ and $P_2$, and we denote this by $P_1 \rightleftharpoons^\ell P_2$. We write $P_1 \rightleftharpoons P_2$ if $P_1 \rightleftharpoons^\ell P_2$ for some $\ell > 0$. If $P_1 \rightleftharpoons^1 P_2$, we say that there is a single conflict and if $P_1 \rightleftharpoons^4 P_2$, we say that there is a total conflict between $P_1$ and $P_4$.*

*Moreover, we define $\mathcal{C}^{\mathcal{A}} = \{(P_i, P_j) \mid P_i, P_j \in \mathcal{A}, P_i \rightleftharpoons P_j\}$ and $\mathcal{C}^{\mathcal{A}}_\ell = \{(P_i, P_j) \mid P_i, P_j \in \mathcal{A}, P_i \rightleftharpoons^\ell P_j\}$.*

By establishing that between any two pairs $P_1$ and $P_2$ of complementary focal elements, there is either a single conflict or there is a total conflict, we can characterize the existence of a hierarchy in terms of single conflicts.

**Lemma$^\star$ 5.2.** *For each frame $\Theta$ and each set $\mathcal{A}$ of pairs of complementary focal elements, $\mathcal{C}^{\mathcal{A}} = \mathcal{C}^{\mathcal{A}}_1 \cup \mathcal{C}^{\mathcal{A}}_4$.*

**Proposition 5.3.** *Let $\Theta$ be a frame of discernment, and let $\mathcal{A}$ be a set with pairs $P_i = (B_i, \overline{B}_i)$ of complementary sets over $\Theta$. Then there exists a hierarchy containing exactly one of $B_i$ and $\overline{B}_i$ for each pair $P_i$ if and only if $\mathcal{C}^{\mathcal{A}} = \mathcal{C}^{\mathcal{A}}_1$.*

*Proof.* Firstly, suppose that $\mathcal{C}^{\mathcal{A}} = \mathcal{C}^{\mathcal{A}}_1$. We argue that the formula $\varphi$ from Theorem 4.2 is satisfied by the following truth assignment $\alpha$, which suffices to show the existence of a suitable hierarchy. For each $i$, let $\alpha(x_i) = 1$ and $\alpha(y_i) = 0$ if $|B_i| \leq |\overline{B}_i|$, and let $\alpha(x_i) = 0$ and $\alpha(y_i) = 1$ otherwise. For space reasons, we omit a detailed proof of this.

Conversely, suppose that there exists a suitable hierarchy. Then the 2CNF formula $\varphi$ from Theorem 4.2 is satisfiable. We argue that no pair $(P_i, P_j)$ can belong to $\mathcal{C}^{\mathcal{A}}_4$. To derive a contradiction, suppose that this were the case, and that $P_i = (B_i, \overline{B}_i)$ and $P_j = (B_j, \overline{B}_j)$. Then, by construction, $\varphi$ would contain the following clauses: $(\nu(B_i) \vee \nu(\overline{B}_i))$, $(\neg\nu(B_i) \vee \neg\nu(\overline{B}_i))$, $(\nu(B_j) \vee \nu(\overline{B}_j))$, $(\neg\nu(B_j) \vee \neg\nu(\overline{B}_j))$, $(\neg\nu(B_i) \vee \neg\nu(B_j))$, $(\neg\nu(B_i) \vee \neg\nu(\overline{B}_j))$, $(\neg\nu(B_j) \vee \neg\nu(B_i))$, and $(\neg\nu(B_j) \vee \neg\nu(\overline{B}_i))$. Thus $\varphi$ would be unsatisfiable, which contradicts Theorem 4.2. Then, one can show $\varphi$ to be unsatisfiable, contradicting Theorem 4.2. By Lemma 5.3, we then know that $\mathcal{C}^{\mathcal{A}} = \mathcal{C}^{\mathcal{A}}_1$. $\square$

## 5.2 MAXIMIZING THE SIZE OF A HIERARCHY

In this section, we study the problem of finding partial hierarchies—among a given set $\mathcal{A}$ of pairs of complementary focal elements—that are as large as possible. In particular, we will show that this problem is closely related to the classical problem VERTEX-COVER, that consists of deciding if a graph has a vertex cover of a given size. Concretely, we show that there is a polynomial-time reduction from VERTEX-COVER to the problem of finding a partial hierarchy of a given site—showing that the latter problem is NP-hard—and also that there is a polynomial-time reduction in the other direction—allowing fixed-parameter tractable algorithms for VERTEX-COVER to be employed for finding hierarchies.

We start with giving a formal definition for the (decision) problem of finding large partial hierarchies, and showing that this problem is NP-complete.

PARTIAL-HIERARCHY
*Input:* A frame $\Theta$ of discernment, a set $\mathcal{A} = \{(B_i, \overline{B}_i)\}_{i=1}^m$ of complementary pairs of focal elements over $\Theta$, and a positive integer $\ell \in \mathbb{N}$.
*Question:* Is there a hierarchy $\mathcal{H} \subseteq \{B_i, \overline{B}_i \mid 1 \leq i \leq m\}$ of size at least $\ell$, such that $\mathcal{H} \cap \{B_i, \overline{B}_i\} \leq 1$ for each $i$?

**Theorem 5.4.** PARTIAL-HIERARCHY *is* NP-*complete.*

*Proof (sketch).* It is straightforward to show that the problem is in NP, and we omit further details on this. To

show NP-hardness, we give a reduction from VERTEX-COVER. Let $G = (V, E)$ be an undirected graph with $V = \{v_1, \ldots, v_m\}$, and let $k \in \mathbb{N}$. We construct an instance of PARTIAL-HIERARCHY. We let $\Theta = \{\star\} \cup V \cup E$. Moreover, we define $\mathcal{A} = \{(B_i, \overline{B}_i)\}_{i=1}^m$ by letting $B_i = \{v_i\} \cup \{e \in E \mid v_i \in e\}$ and $\overline{B}_i = \Theta \setminus B_i$. Finally, we let $\ell = m - k$.

Then $G$ has a vertex cover of size $k$ if and only if there is a hierarchy $\mathcal{H} \subseteq \{B_i, \overline{B}_i \mid 1 \le i \le m\}$ of size $\ell$. In particular, for any $C \subseteq V$ it holds that $C$ is a vertex cover of $G$ if and only if $\mathcal{A}_C = \{(B_i, \overline{B}_i) \mid v_i \in V \setminus C\}$ admits a hierarchy (in the sense of Theorem 4.2).

To show this, the following claim is central. For each $v_i, v_j \in V$ such that $i \neq j$, if $\{v_i, v_j\} \in E$, then $(B_i, \overline{B}_i) \rightleftharpoons^{\text{4}} (B_j, \overline{B}_j)$, and if $\{v_i, v_j\} \notin E$, then $(B_i, \overline{B}_i) \rightleftharpoons^{\text{1}} (B_j, \overline{B}_j)$. The above correspondence between vertex covers $C$ of $G$ and partial hierarchies $\mathcal{A}_C$ is straightforward to show, using this claim and Proposition 5.3. $\qquad\square$

**Proposition⋆ 5.5.** *There is a polynomial-time reduction from* PARTIAL-HIERARCHY *to* VERTEX-COVER *that maps instances* $(\Theta, \mathcal{A}, \ell)$ *to instances with* $k = |\mathcal{A}| - \ell$.

The result of Proposition 5.5 shows that we can use fixed-parameter tractable algorithms for VERTEX-COVER to find partial hierarchies, and that such algorithms can be expected to run efficiently in cases where we can obtain a hierarchy from $\mathcal{A}$ by removing only few items. In particular, we can find vertex covers of size $k$ in time $O(1.2738^k + kn)$ [Chen et al., 2010], which is a running time that is manageable whenever $k = m - \ell$ is reasonably small. Additionally, one could employ approximation algorithms for finding minimum-size vertex covers (see, e.g., Arora and Barak, 2009) to get partial hierarchies that approximate those of maximum size.

# 6 USING HIERARCHICAL STRUCTURE IN ARBITRARY BODIES OF EVIDENCE

In Sections 4 and 5 we studied the problem of determining in what situations—possibly after disregarding some pieces of evidence—evidence is entirely aligned with a hierarchy. Of course, there are also situations where this is not the case (and where disregarding evidence is undesirable or inappropriate). In this section, we take some initial steps towards algorithmically using hierarchical structure to combine evidence also in cases where the evidence is not entirely in line with any hierarchy.

In particular, we introduce a measure of how much (of a particular type of) hierarchical structure there is in any set $\mathcal{A}$ of focal elements, and we give an algorithm to compute the combined belief of a given set—based on applying DRC

to simple support functions with focal elements in $\mathcal{A}$—that works efficiently when there is a high degree of hierarchical structure in $\mathcal{A}$.

The main idea behind this measure (and the algorithm) is as follows. Whenever there are focal elements $A_1, A_2 \in \mathcal{A}$ that are conflicting—in the sense of Definition 5.1—we merge them together. We do this merging iteratively until there are no conflicts remaining, and thus until we have a hierarchy. The algorithm, roughly, works in a two-step fashion: first, (i) for all focal elements in the hierarchy that are the result of such a merging operation, we compute the combined belief using a brute-force algorithm; then, (ii) we use the algorithm of Theorem 3.6 to combine these intermediate results—for the merged focal elements—with the evidence for focal elements that are not the result of any merging operation.

Step (i) of this algorithm takes exponential time, but this is only exponential in the size of the merged focal elements. As measure of the amount of hierarchical structure in the set $\mathcal{A}$ of focal elements we take the size $k$ of the largest focal element resulting from this iterative merging process. The smaller this number $k$, the more hierarchical structure the set $\mathcal{A}$ contains, and in fact, if $\mathcal{A}$ is already a hierarchy, then $k = 0$. The running time of the algorithm then is $2^k \cdot \text{poly}(|x|)$, where $x$ denotes the size of the problem input. In other words, the algorithm runs in *fixed-parameter tractable time*, when we consider as parameter the amount of hierarchical structure.

## 6.1 MEASURING HIERARCHICAL STRUCTURE

Let us now work out this idea in more detail, and let us begin with the measure of the amount of hierarchical structure in any given set of focal elements.

**Definition 6.1** (Corresponding merged hierarchy). *Let $\Theta$ be a frame of discernment and let $\mathcal{A} = \{A_1, \ldots, A_m\}$ be a set of focal elements $A_i \subseteq \Theta$. Then the merged hierarchy $\mathcal{H}_\mathcal{A}$ corresponding to $\mathcal{A}$ is defined by the following procedure. Initially, let $\mathcal{A}_{origin} = \mathcal{A}$ and $\mathcal{A}_{merged} = \emptyset$, and then iteratively update $(\mathcal{A}_{origin}, \mathcal{A}_{merged})$ using the following rules until no rule applies anymore.*

- *If there are $A_i, A_j \in \mathcal{A}_{origin}$ such that both (i) $A_i \cap A_j \neq \emptyset$ and (ii) neither $A_i \subseteq A_j$ nor $A_j \subseteq A_i$, then replace $\mathcal{A}_{origin}$ by $\mathcal{A}_{origin} \setminus \{A_i, A_j\}$ and add $A_i \cup A_j$ to $\mathcal{A}_{merged}$.*

- *If there is some $A_i \in \mathcal{A}_{origin}$ and some $A_j \in \mathcal{A}_{merged}$ such that $A_i \cap A_j \neq \emptyset$, then replace $\mathcal{A}_{origin}$ by $\mathcal{A}_{origin} \setminus \{A_i\}$ and replace $\mathcal{A}_{merged}$ by $\mathcal{A}_{merged} \setminus \{A_j\} \cup \{A_i \cup A_j\}$.*

- *If there are $A_i, A_j \in \mathcal{A}_{merged}$ such that $A_i \cap A_j \neq \emptyset$, then replace $\mathcal{A}_{merged}$ by $\mathcal{A}_{merged} \setminus \{A_i, A_j\} \cup \{A_i \cup A_j\}$.*

*Finally, let $\mathcal{H}_\mathcal{A} = \mathcal{A}_{origin} \cup \mathcal{A}_{merged}$.*

If $\mathcal{A}$ is already a hierarchy, then none of these rules ever applies, and thus $\mathcal{H}_\mathcal{A} = \mathcal{A}$.

No matter in which order you apply the rules in this iterative procedure, the result does not change. In other words, for any $\mathcal{A}$, the hierarchy $\mathcal{H}_\mathcal{A}$ is uniquely defined.

**Proposition 6.2.** *For each set $\mathcal{A}$ of focal elements, the procedure in Definition 6.1 yields a unique $\mathcal{H}_\mathcal{A}$, regardless of the order in which rules are applied. Moreover, $\mathcal{H}_\mathcal{A}$ is a hierarchy, and for each $A \in \mathcal{A}$ there is some $H \in \mathcal{H}_\mathcal{A}$ such that $A \subseteq H$.*

*Proof (sketch).* Each of the rules only merges sets, which directly gives us termination and the property that for each $A \in \mathcal{A}$ there is some $H \in \mathcal{H}_\mathcal{A}$ such that $A \subseteq H$. If the resulting $\mathcal{H}_\mathcal{A}$ were not a hierarchy, then one could still apply a rule, which proves that $\mathcal{H}_\mathcal{A}$ must be a hierarchy. Uniqueness can be proved with the observation that the effects of the rules only strictly increase the sets in $\mathcal{A}_{\text{merged}}$, and the preconditions of the rules are monotone—in the sense that making sets in $\mathcal{A}_{\text{merged}}$ larger will not make a previously applicable rule not applicable anymore. $\square$

Having the notion of corresponding merged hierarchies in place, we introduce the level of merging needed to go from $\mathcal{A}$ to $\mathcal{H}_\mathcal{A}$ as a way to measure the amount of hierarchical structure in $\mathcal{A}$.

**Definition 6.3** (Level of merging). *Let $\Theta$ be a frame of discernment, and let $\mathcal{A} = \{A_1, \ldots, A_m\}$ be a set of focal elements $A_i \subseteq \Theta$. We define the* level of merging *needed to go from $\mathcal{A}$ to its corresponding merged hierarchy $\mathcal{H}_\mathcal{A}$ to be $k = \max_{A \in \mathcal{A}_{merged}} |A|$, where $\mathcal{A}_{origin}$ and $\mathcal{A}_{merged}$ are given by the procedure described in Definition 6.1.*

The procedure described in Definition 6.1 gives us a polynomial-time algorithm to compute both $\mathcal{H}_\mathcal{A}$ and $k$.

**Proposition 6.4.** *For each $\mathcal{A}$, we can in polynomial time compute its corresponding hierarchy $\mathcal{H}_\mathcal{A}$ and compute the level $k$ of merging needed to go from $\mathcal{A}$ to $\mathcal{H}_\mathcal{A}$.*

*Proof (sketch).* The procedure described in Definition 6.1 terminates in polynomial time and produces $\mathcal{H}_\mathcal{A}$ and $k$. $\square$

**Example 6.5.** *Consider $\Theta = \{a, b, c, d, e\}$ and $\mathcal{A} = \{\{a\}, \{a, b\}, \{b, c\}, \{a, b, c, d\}, \{d\}, \{e\}\}$. Then $\mathcal{H}_\mathcal{A} = \{\{a, b, c\}, \{a, b, c, d\}, \{d\}, \{e\}\}$ and the level $k$ of merging needed to go from $\mathcal{A}$ to $\mathcal{H}_\mathcal{A}$ is 3, as $\{a, b, c\}$ is the largest element in the set $\mathcal{A}_{merged}$ resulting from the procedure described in Definition 6.1.*

## 6.2 USING HIERARCHICAL STRUCTURE

We now have everything in place to present our fixed-parameter tractable algorithm that extends the result of Shafer and Logan [2008] (Theorem 3.6) to arbitrary sets $\mathcal{A}$ of focal elements.

**Theorem 6.6.** *Let $\Theta$ be a frame of discernment and let $\mathcal{A} = \{A_1, \ldots, A_m\}$ be a set of focal elements $A_i \subseteq \Theta$. Moreover, let $\mathcal{H}_\mathcal{A}$ be the hierarchy corresponding to $\mathcal{A}$, and let $k$ be the level of merging needed to go from $\mathcal{A}$ to $\mathcal{H}_\mathcal{A}$.*

*Then, given simple b.p.a.'s $m_1, \ldots, m_n$, each of whose single proper focal element is an element of $\mathcal{A}$, for each $A \in \mathcal{A}$ we can compute $\text{Bel}_m(A)$, $\text{Bel}_m(\overline{A})$, $\text{Plau}_m(A)$ and $\text{Plau}_m(\overline{A})$, for $m = \bigoplus_{i=1}^n m_i$, in time $2^k \cdot poly(|x|)$ where $x$ denotes the problem input.*

*Proof (sketch).* We describe how to compute $\text{Bel}_m(A)$. This procedure can be straightforwardly modified to compute $\text{Bel}_m(\overline{A})$, $\text{Plau}_m(A)$ and $\text{Plau}_m(\overline{A})$ as well. We may assume without loss of generality, that for each $A \in \mathcal{A}$, there is exactly one b.p.a. among $m_1, \ldots, m_n$ that has $A$ as proper focal element—call this b.p.a. $m_A$.

We will use the following procedure. Firstly, we construct $\mathcal{H}_\mathcal{A}$, together with $\mathcal{A}_{\text{origin}}$ and $\mathcal{A}_{\text{merged}}$, as described in Definition 6.1. Then, for each $A \in \mathcal{A}_{\text{merged}}$, we use a brute-force approach to compute $m_A = \bigoplus_{A' \in \mathcal{A}, A' \subseteq A} m_{A'}$, and we construct the simple b.p.a. $m'_A$ with proper focal element $A$ such that $m'_A(A) = \text{Bel}_{m_A}(A)$. Then we use Theorem 3.6, using $m_A$ for each $A \in \mathcal{A}_{\text{origin}}$ and $m'_A$ for each $A \in \mathcal{A}_{\text{merged}}$, to compute $\text{Bel}_m(H)$ for each $H \in \mathcal{H}_\mathcal{A}$. By Lemma 6.7, we can safely replace $m_A$ by $m'_A$ in this computation, for each $A \in \mathcal{A}_{\text{merged}}$. What remains is to compute $\text{Bel}_m(A)$ for each $A \in \mathcal{A} \setminus \mathcal{H}_\mathcal{A}$. Lemma 6.8 gives us a direct way to do this using values that we have already computed.

The computation of $m(A)$ and $m'(A)$ for $A \in \mathcal{A}_{\text{merged}}$ can be done in time $2^k \cdot \text{poly}(|x|)$. Moreover, given $m(A)$ and $m'(A)$ for each $A \in \mathcal{A}_{\text{merged}}$, the remainder of the algorithm can be carried out in polynomial time.

$\square$

The proof of Theorem 6.6 uses the following two lemmas, which can be established straightforwardly using the definition of Dempster's rule of combination.

**Lemma 6.7.** *Let $\Theta$ be a frame of discernment, let $m_1, m_2, m_3$ be b.p.a.'s over $\Theta$, and let $A \subseteq \Theta$ be such that $\text{Bel}_{m_1}(A) + m_1(\Theta) = 1$, $\text{Bel}_{m_2}(A) + m_2(\Theta) = 1$, and $\text{Bel}_{m_3}(\Theta \setminus A) + m_3(\Theta) = 1$. Moreover, let $m_{1,3} = m_1 \oplus m_3$ and let $m_{2,3} = m_2 \oplus m_3$. Then $\text{Bel}_{m_{1,3}}(A) = \text{Bel}_{m_{2,3}}(A)$ and $\text{Bel}_{m_{1,3}}(B) = \text{Bel}_{m_{2,3}}(B)$ for each $B \subseteq \Theta \setminus A$.*

**Lemma 6.8.** *Let $\Theta$ be a frame of discernment, let $m_1, m_2$ be b.p.a.'s over $\Theta$, and let $A \subseteq \Theta$ be such that $\text{Bel}_{m_1}(A) + m_1(\Theta) = 1$, and $\text{Bel}_{m_2}(\Theta \setminus A) + m_2(\Theta) = 1$. Then $\text{Bel}_{m_1}(B)/\text{Bel}_{m_1}(A) = \text{Bel}_{m_{1,2}}(B)/\text{Bel}_{m_{1,2}}(A)$ for each $B \subseteq A$, where $m_{1,2} = m_1 \oplus m_2$.*

## 6.3 EXTENSIONS

The result of Theorem 6.6 provides a starting point for investigating how best to algorithmically use hierarchical structure to combine arbitrary sets of evidence. By itself, the result is restricted in various ways. In this section, we discuss several suggestions for how to extend Theorem 6.6 to more general and practically useful settings.

**Including complementary focal elements**   The algorithm of Theorem 6.6 can straightforwardly be adapted also to the case where you additionally have simple b.p.a.'s whose focal element is the complement of some $A \in \mathcal{H}_\mathcal{A}$—as is the case for Theorem 3.6. Therefore, one might be able to compute $\mathrm{Bel}_m(A)$ for some $A \in \mathcal{A}$ more efficiently by constructing a set $\mathcal{A}' \subseteq \mathcal{A}$ such that for each focal element $D$ of a given simple b.p.a., either (i) $\mathcal{H}_{\mathcal{A}'}$ contains a set in $\mathcal{A}_{\mathrm{merged}}$ that is a superset of $D$, or (ii) $D$ is the complement of some set in $\mathcal{H}_{\mathcal{A}'}$.

For example, take $\mathcal{A} = \{\{a, b\}, \{b, c\}, \{a, b, c, d\}, \{d, e\}, \{e\}\}$, and suppose you have simple b.p.a.'s with proper focal elements in $\mathcal{A}$. Then $\mathcal{H}_\mathcal{A} = \{\{a, b, c, d, e\}\}$. However, you can also consider the hierarchy $\mathcal{H} = \{\{a, b, c\}, \{a, b, c, d\}, \{e\}\}$ to use (the extended variant) of the algorithm of Theorem 6.6 to compute $\mathrm{Bel}_m(A)$ for sets $A \in \mathcal{A}$, and this would be more efficient. An interesting direction for future research would be to develop (efficient) algorithms for finding a set $\mathcal{A}' \subseteq \mathcal{A}$ that enables the most efficient use of the algorithm of Theorem 6.6.

**Improved distance measures**   The notion of a merged hierarchy $\mathcal{H}_\mathcal{A}$ corresponding to a set $\mathcal{A}$ of focal elements is conceivable that in many situations one is forced to merge (nearly) all sets in $\mathcal{A}$. In such situations, the algorithm of Theorem 6.6 would boil down to combining all available evidence using a brute force algorithm.

It would be interesting to study more refined notions of distance to a hierarchy. One possible approach for finding a more fine-grained notion of distance that could strike such a balance would be to count the number of steps needed to go from an arbitrary set $\mathcal{A}$ of focal elements to a hierarchy $\mathcal{H}$ using operations on the sets in $\mathcal{A}$ that do not force one to use a brute force algorithm to deal with the resulting sets.

## 7   CONCLUSION

We have presented different solutions for combining pieces of evidences through Dempster's rule of combination in polynomial (or fixed-parameter tractable) time. We showed that one can check in polynomial time if a given set of evidence admits a hierarchy, in which case one can combine evidence in polynomial time. For the non-hierarchical case, we considered two variants of the problem of finding a partial hierarchy, and showed that both variants are NP-hard

but admit fixed-parameter tractable algorithm (where the parameter is the number of sets to delete). In addition, we provided a fixed-parameter tractable algorithm for combining evidence where the parameter in some sense measures how hierarchical the body of evidence is. Our work points to several interesting open problems. One example is to study how filtering the original set of evidence (e.g., to get a hierarchy structure) affects the final result and how this relates to the notion of relevance for evidence. Another direction would be to explore algorithms that use hierarchical structure in arbitrary sets of evidence in a more efficient way.

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
