# OpenReview forum: "Using hierarchies to efficiently combine evidence with Dempster's rule of combination"
_auai.org/UAI/2022/Conference — UAI 2022 Poster_

### Official Review · Reviewer_p71b · 2022-04-01

**Q2(1) Originality/Novelty:** 3
**Q2(2) Significance/Impact:** 2
**Q2(3) Correctness/Technical Quality:** 4
**Q2(6) Clarity Of Writing:** 4
**Q6 Overall Score:** 8
**Q8 Confidence In Your Score:** 4

**Q1 Summary And Contributions:**

The authors study the complexity of Dempster's rule, exploring various restrictions to the problem for which lower complexity can be achieved. They focus on the case of evidence represented in hierarchical form, providing both algorithms to check whether a body of evidence conforms to a given hierarchy, or to transform bodies of evidence into a relaxed hierarchy, and provide algorithms (in polynomial or fixed-parameter polynomial time) for computing the combination of such bodies of evidence.

**Q2 Assessment Of The Paper:**

More detailed information regarding each of these aspects is given below:

**Q2(5) Reproducibility:**

3: Good: Key resources (e.g., proofs, code, data) are available and key details (e.g., proofs, experimental setup) are sufficiently well-described for competent researchers to confidently reproduce the main results.

**Q3 Main Strengths:**

- The paper is really well written and clear
- The results are interesting and well presented
- The importance of the results in the given domain is well discussed
- Most proofs are clearly written and easy to follow

**Q4 Main Weakness:**

- Some proofs are too sketched and would be better presented by expanding them in an appendix

**Q5 Detailed Comments To The Authors:**

The authors study the complexity of Dempster's rule, starting from the well-known fact
that the problem is #P-complete and exploring various restrictions to the problem for
which lower complexity can be achieved. In particular, they focus on the case
of evidence represented in hierearchical form, providing both algorithm to check
whether a body of evidence conforms to a given hierarchy, or to transform bodies of
evidence in a relaxed hierarchy, and provide algorithms (in polynomial or fixed-parameter polynomial time) for computing the combination of such bodies of evidence.

The paper is very well written, the exposition is good and the proofs are clear, even if some proof sketches are almost too sketched (all results seem to be correct by working out the proof by onself, however maybe the authors could add additional details in an Appendix, e.g. for Theorem 5.4 and Proposition 6.4).

The results are well-motivated, and it is clear that from a theoretical point of view this work expands on the state-of-the-art of algorithms for Dempster-Shafer theory, at least to my knowledge. Aside from the theoretical point of view, however, I think it would be interesting if the authors could provide some indications in regard to what degree the proposed algorithms allow to improve over naive application of Dempster's combination rule on some examples or benchmarks: this would be particularly interesting for the fixed-parameter tractable algorithms, as in some cases (e.g., when in most practical problems the size of the parameter is relatively large) such algorithms fail to offer a significant advantage compared with the more naive solutions (and often with a non-trivial increase in implementation complexity).

**Q7 Justification For Your Score:**

The paper provides a substantial improvement in the theoretical knowledge about the computational complexity of relevant problems in Dempster-Shafer theory, an important approach for managing uncertainty and evidence. The paper is well written, and the technical part (theorems, proof) seems to be correct after trying to replicate the proofs. The content and its relevance is well motivated, and the limitations of the proposed approach are discussed with honesty in the conclusion of the paper.

**Q9 Complying With Reviewing Instructions:**

1: Yes.

---

### Official Review · Reviewer_JtE3 · 2022-04-11

**Q2(1) Originality/Novelty:** 3
**Q2(2) Significance/Impact:** 2
**Q2(3) Correctness/Technical Quality:** 3
**Q2(6) Clarity Of Writing:** 3
**Q6 Overall Score:** 6
**Q8 Confidence In Your Score:** 4

**Q1 Summary And Contributions:**

 The paper characterizes situations where the Dempster combination rule can be done in polynomial time.


**Q2 Assessment Of The Paper:**

More detailed information regarding each of these aspects is given below:

**Q2(5) Reproducibility:**

3: Good: Key resources (e.g., proofs, code, data) are available and key details (e.g., proofs, experimental setup) are sufficiently well-described for competent researchers to confidently reproduce the main results.

**Q3 Main Strengths:**

New contributions on the computational complexity of Dempster combination rule with identification of situation where tractable methods can be identified.

**Q4 Main Weakness:**

Definitions are dense. Results need to be more justified.

**Q5 Detailed Comments To The Authors:**

Results of this paper are proposed in the context of belief functions for representing and reasoning with uncertain information.

The authors are interested in one of the fundamental operator in belief functions: Demspter's conjunctive combination rule.

More precisely, the paper characterizes situations where the Dempster combination rule can be done in polynomial time.

The authors focus on the combination of simple support functions (with a single focal element).

The key to the paper can be found in the result of Shafer and Logan [2008], where a polynomial algorithm for calculating plausibility functions; obtained by combination by Dempster's rule of simple support functions, is provided. The tractability became possible if the focal elements follow a hierarchical structure.

Starting from this result, the authors search under which conditions such hierarchies exist and therefore a polynomial algorithm for Dempster's rule can be found.

The paper is generally well written (even if some definitions are dense). One can follow the main results of the paper. It has more or less significant contributions, a number of complexity results with reductions of problems.

A few remarks:
-- The example given on page 1 is not re-used in the paper; whereas it was expected to be a running example of the paper.

-- the restriction to the simple support function announced on page 2 is not justified (we understand this later). We expect a little more motivation. In particular, the combination of two simple support functions is not a simple support function. So at this level, we expect explanations why consider this restrictive framework.

-- -- I propose to give the definition of the rule of combination with simple support functions. There are interesting properties that may explain the interest of restricting to simple support functions

-- -- In theorem 4, it is not clear whether the (dichotomous) restrictions limit the significance and importance of the theorem.

-- Finally, I find that an important reference is missing (P. Smets (1995). The canonical
decomposition of a weighted belief. In
Proceedings of the Fourteenth
International Joint Conference on
Artificial Intelligence, volume 2, pages
1896−1901, Montreal, Canada, August
1995) which concerns the decomposition of mass function into simple support function.

**Q7 Justification For Your Score:**

There are new contributions that justify my score.

**Q9 Complying With Reviewing Instructions:**

1: Yes.

---

### Official Review · Reviewer_KM9m · 2022-04-11

**Q2(1) Originality/Novelty:** 3
**Q2(2) Significance/Impact:** 3
**Q2(3) Correctness/Technical Quality:** 3
**Q2(6) Clarity Of Writing:** 3
**Q6 Overall Score:** 6
**Q8 Confidence In Your Score:** 3

**Q1 Summary And Contributions:**

This papers proposes an approach to merge bpa's (actually support functions and dichotomous bpa) by searching for a hierarchy of evidences - and then applying an algorithm due to Shafer and Logan. It also shows that the problem addressed is difficult in the general case (#P complete)

**Q2 Assessment Of The Paper:**

More detailed information regarding each of these aspects is given below:

**Q2(4) Quality Of Experiments (Optional):**

1: Poor: The experimental evaluation is flawed or the results fail to adequately support the main claims.

**Q2(5) Reproducibility:**

3: Good: Key resources (e.g., proofs, code, data) are available and key details (e.g., proofs, experimental setup) are sufficiently well-described for competent researchers to confidently reproduce the main results.

**Q3 Main Strengths:**

* the complexity analysis
* the clear writing
* the  coherence of the approach (polytime algorithm if possible, fixed parameter tractable algorithm if not)

**Q4 Main Weakness:**

*  no experimental study is provided

**Q5 Detailed Comments To The Authors:**

As far as I understand, the source of complexity is, in the present study, the number of bpa's  to merge.  But in application cited as motivation (diagnosis, combination of human opinion), this number is generally low. What are the other sources of complexity ?

An experimental study on real problems (thus, with a low number of bpa to merge) should be easy to conduct ...such a study would make the paper complete.

What about a generalization of the result to mass functions containing disjoint evidences (the all the focal elements of a given bpa are disjoint) - to what extent the number of such "basic evidences"  in each bpa is a source of complexity ?

**Q7 Justification For Your Score:**

I appreciate the writing of the paper (motivation, clarity, structure and  logic of the presentation, from complexity analysis to algorithms).

The problem addressed is not trivial and the domain is missing such algorithmic and complexity studies - it is important to provide the domain with efficient tools.

The results are convincing from a theoretical point of view, and should be generalized to more complex bpas

Nevertheless, the results are still to be supported by an experimental study.

**Q9 Complying With Reviewing Instructions:**

1: Yes.

---

### Official Review · Reviewer_NA8U · 2022-04-13

**Q2(1) Originality/Novelty:** 3
**Q2(2) Significance/Impact:** 3
**Q2(3) Correctness/Technical Quality:** 3
**Q2(6) Clarity Of Writing:** 4
**Q6 Overall Score:** 8
**Q8 Confidence In Your Score:** 4

**Q1 Summary And Contributions:**

The paper provides an in-depth investigation of the computational complexity of applying Dempster's Rule of Cimbination to combine evidence in the special case where the set of evidence has a hierarchical shape.

**Q2 Assessment Of The Paper:**

More detailed information regarding each of these aspects is given below:

**Q2(5) Reproducibility:**

4: Excellent: Key resources (e.g., proofs, code, data) are available and key details (e.g., proof sketches, experimental setup) are comprehensively described for competent researchers to confidently and easily reproduce the main results.

**Q3 Main Strengths:**

This paper casts some light on the complexity of evidence combination, which is little explored in the literature, characterizing some cases in which the complexity, which in general is #P-hard, can dramatically drop to polynomial. That has important consequences for the scalability of Dempster-Shafer Theory.

**Q4 Main Weakness:**

Cannot see any

**Q5 Detailed Comments To The Authors:**

This is an extremely interesting contribution, which, I hope, will pave the ways to more investigations into the characterization of complexity of evidence combination. The paper is very well written and technically sound to the extent that I could check (I didn't check the proofs in the appendix).

**Q7 Justification For Your Score:**

Very interesting and solid contribution, fully relevant to the main topic of this conference.

**Q9 Complying With Reviewing Instructions:**

1: Yes.

---

### Decision · Program_Chairs · 2022-05-15

**Decision:**

Accept (Poster)

**Comment:**

Meta Review: Pros:
1. The paper makes non-trivial advances over the current state-of-the-art for hierarchical models.
2. The paper is likely to have high impact within a subfield of AI OR moderate impact across more than one subfield of AI.
3. Key resources (e.g., proofs, code, data) are available and key details (e.g., proof sketches, experimental setup) are comprehensively described for competent researchers to confidently and easily reproduce the main results.
4. The paper is well-organized and clearly written.

Cons:
1. Some proofs are too sketched and would be better presented by expanding them in an appendix
2. Some definitions are dense. Results need to be more justified.